# Increased O-GlcNAcylation in Leukocytes from Overweight Pediatric Subjects: A Pilot Study

**DOI:** 10.3390/ijms26125665

**Published:** 2025-06-13

**Authors:** Alessia Remigante, Sara Spinelli, Gianluca Rizzo, Daniele Caruso, Angela Marino, Elisabetta Straface, Silvia Dossena, Rossana Morabito

**Affiliations:** 1Department of Biomedical, Dental and Morphological and Functional Imaging, University of Messina, 98125 Messina, Italy; 2Department of Chemical, Biological, Pharmaceutical and Environmental Sciences, University of Messina, 98125 Messina, Italy; saspinelli@unime.it (S.S.); marinoa@unime.it (A.M.); rmorabito@unime.it (R.M.); 3Independent Researcher, 98121 Messina, Italy; drgianlucarizzo@gmail.com; 4Complex Operational Unit of Clinical Pathology of Papardo Hospital, 98166 Messina, Italy; caruso.daniele1985@libero.it; 5Biomarkers Unit, Center for Gender-Specific Medicine, Istituto Superiore di Sanità, 00161 Rome, Italy; elisabetta.straface@iss.it; 6Institute of Pharmacology and Toxicology, Research and Innovation Center Regenerative Medicine & Novel Therapies, Paracelsus Medical University, 5020 Salzburg, Austria; silvia.dossena@pmu.ac.at

**Keywords:** O-GlcNAcylation, leukocytes, pediatric overweight, type II diabetes mellitus, pre-diabetes

## Abstract

Type II diabetes mellitus (T2D) is a metabolic disorder. Childhood overweight or obesity raises the risk for developing T2D later in life. Early identification of at-risk individuals is fundamental for disease prevention and patient management. The scope of this pilot study was to explore whether leukocyte protein O-GlcNAc modification is elevated in an overweight pediatric cohort. Eight overweight and eight normal-weight children aged 3–13 years were recruited at the Papardo General Hospital (Messina, Italy). Physical exams, complete blood tests, and determination of leukocyte protein O-GlcNAcylation were carried out. Protein O-GlcNAcylation was higher in leucocytes from overweight children compared to normal-weight children, and was significantly correlated with BMI, metabolic markers (LDL-cholesterol/triglycerides), and the inflammatory marker CRP. This study suggests that leukocyte protein O-GlcNAcylation may represent a novel biomarker for the early detection of metabolic abnormalities that may lead to the development of pre-diabetes or T2D later in life.

## 1. Introduction

Obesity in children and adolescents is an alarming worldwide phenomenon. The World Health Organization (WHO) estimated that over 390 million children and adolescents aged 5–19 years were overweight (BMI-for-age greater than 1 standard deviation above the WHO Growth Reference median) in 2022, of whom 160 million were obese (BMI-for-age greater than 2 standard deviations above the WHO Growth Reference median) [1]. Among EU countries, Italy has a relatively high childhood overweight prevalence (>20%) [2]. Based on the results of national surveillance conducted in 2023 on a sample of >46,000 school-age children, the prevalence of overweight increases significantly from North to South Italy, with high figures in Sicily (20.5%) [3]. Overweight is the most important risk factor for Type II diabetes mellitus (T2D) development in youth [4,5,6]. The onset of diabetes is preceded by a condition of silent insulin resistance and impaired glucose tolerance that can last for over a decade. Notable efforts have been made by WHO in compiling and revising guidelines for T2D. Nevertheless, a substantial number of T2D cases remain undiagnosed, and up to 35% of adult patients already present severe complications at diagnosis—clearly accounting for the late detection of the disease [7]. Undiagnosed T2D and pre-diabetes are also prevalent among children and adolescents [8,9]. Evidence exists for faster progression of T2D and related complications in pediatric T2D patients compared to Type I diabetes mellitus (T1D) or adult-onset T2D [10]. Therefore, there is a great need for new molecular biomarkers for the identification of at-risk individuals and the early detection of the disease.

O-GlcNAcylation is a post-translational modification consisting of the addition of a carbohydrate (N-acetylglucosamine, GlcNAc) on hydroxyl groups of serine and/or threonine residues of cellular proteins. Approximately 1–3% of glucose molecules entering the cell are diverted to the hexosamine biosynthetic pathway, which is therefore upregulated in hyperglycemia and is considered a sensor of the metabolic status of the cell [11]. A correlation between protein O-GlcNAcylation abundance and insulin resistance, glucose toxicity, and complications of T2D has been firmly established [12,13]. O-GlcNAcylation is differentially regulated at specific sites on erythrocyte proteins in response to glycemic status [14]. Protein O-GlcNAc modification is catalyzed by the enzymes O-GlcNAc-transferase (OGT) and O-GlcNAcase (OGA). Park and co-authors found that OGA expression was increased in erythrocytes from individuals with diabetes and pre-diabetes compared to the control population, and therefore suggested that OGA levels could be even superior to glycated hemoglobin (A1c) in detecting pre-diabetes [15]. Springhorn and collaborators have demonstrated that OGA mRNA levels and activity were increased in leukocytes of pre-diabetic and diabetic patients compared to healthy volunteers. The increased OGA expression was interpreted as an adaptive response to hyperglycemia-induced hyper-O-GlcNAcylation [16]. Surprisingly, OGA activity did not correlate with glycemic markers, including A1c, but rather correlated with inflammation markers, such as C-reactive protein (CRP) and mRNA of pro-inflammatory proteins [17]. Additionally, a marked increase in O-GlcNAcylation is also linked to mitochondrial dysfunction, which in turn elevates oxidative stress. O-GlcNAcylation and oxidative stress interact in a reciprocal manner, playing a pivotal role in the metabolic disturbances and insulin resistance [18]. Oxidative stress, in turn, influences the enzymes involved in the O-GlcNAcylation process, leading to changes in protein function and inflammatory responses. This bidirectional interaction may exacerbate cellular dysfunction and contribute to the early pathophysiological changes associated with metabolic diseases [19,20,21]. Although there is a common consensus on the fact that O-GlcNAcylation in blood cells is elevated in T2D, and leukocytes may more readily reflect the physiological status of the organism as they are capable of de novo protein synthesis, whether leukocyte protein O-GlcNAcylation can be used as a predictor of T2D risk is not established. Specifically, there is a lack of studies exploring the O-GlcNAc status in at-risk populations, such as overweight pediatric populations.

The objectives of this single-center pilot study were to (i) demonstrate the feasibility of detecting leukocyte protein O-GlcNAcylation in pediatric subjects; (ii) determine the number of participants to be recruited for a full-scale study; and (iii) obtain preliminary data on whether protein O-GlcNAc modification is elevated in leukocytes of a cohort of excess-weight non-diabetic pediatric subjects compared to normal-weight subjects. The potential correlation of the O-GlcNAc status with metabolic and inflammatory blood parameters was also explored.

## 2. Results

In Figure 1, all normal-weight and overweight female (Figure 1a) and male (Figure 1b) study participants are categorized based on the standard deviation (SD) from the mean BMI distribution of a reference population, according to WHO guidelines [1]. In Table 1, specific clinical–metabolic parameters of these subjects divided by gender groups are also reported. All values fell into the physiological reference range, with the exception of CRP, which was increased in both female and male overweight subjects. No statistically significant differences between the clinical–metabolic parameters of overweight male and female participants were detected; however, insulin, triglycerides, total cholesterol, LDL-cholesterol, and CRP levels tended to be higher in overweight females than in males (Table 1). Figure 1c,d show O-GlcNAc-modified protein expression levels detected in leucocytes obtained from normal-weight and overweight participants, respectively. In overweight subjects, O-GlcNAcylation was significantly higher than that detected in normal-weight subjects. Based on the SD of leukocyte protein O-GlcNAcylation determined in overweight subjects in this pilot study, *n* = 17 participants should be recruited for each study group to detect statistically significant differences with a margin of error of 1% and a confidence level of 95% in a full-scale study.

A partial correlation analysis of overweight subjects accounting for confounders (gender and age for metabolic/inflammatory parameters and white blood cell count for O-GlcNAcylation) revealed significant association between leukocyte protein O-GlcNAcylation and BMI (Figure 1e), lipid metabolism markers (triglycerides and LDL-cholesterol, respectively, Figure 1f,g), and the inflammatory marker CRP (Figure 1h). O-GlcNAcylation status did not significantly correlate with hyperglycemia markers (FPG and A1c). No differences in the O-GlcNAcylation between female and male subjects were observed.

## 3. Discussion

Although there is a common consensus regarding increased O-GlcNAcylation signaling in pre-diabetes and diabetes, there is a paucity of studies exploring whether the O-GlcNAcylation status might identify at-risk individuals prior to overt alterations in metabolic blood parameters occurring. The present investigation was undertaken to determine whether protein O-GlcNAcylation is upregulated in overweight children. First, we observed that protein O-GlcNAcylation was higher in leucocytes of individuals from an excess-weight pediatric cohort compared to age- and population-matched normal-weight children. Noteworthy, in these overweight subjects, A1c values were within the reference range. It is generally acknowledged that A1c levels represent chronic hyperglycemia well but do not sufficiently reflect glycemic instability [22]. Thus, A1c levels correlate weakly with insulin resistance and may fail to detect the early onset of pre-diabetes. FPG and oral glucose tolerance test (OGTT) are deemed superior to A1c in detecting pre-diabetes [23]. Although the OGTT was not performed in this study, FPG levels were normal in our cohort.

Next, the clinical–biological blood parameters of excess-weight participants were tested for correlation with the respective O-GlcNAc levels. An important observation from this study is the significant positive correlation between the abundance of O-GlcNAc-modified proteins and CRP, an inflammatory marker produced by the liver whose expression typically increases in chronic inflammatory conditions such as overweight and T2D [24].

Since former studies found increased O-GlcNAcylation signaling in pre-diabetes and diabetes [14,15,16], we expected a positive correlation between leukocyte protein O-GlcNAcylation and established markers of hyperglycemia. Surprisingly, however, we found no significant correlation between O-GlcNAc levels and FPG, A1c, and insulin. Instead, positive correlations were found between O-GlcNAc-modified protein levels and BMI, triglycerides, and LDL-cholesterol, suggesting that protein O-GlcNAcylation is strongly connected to lipid metabolism. BMI correlates well with fat mass in children, providing a valid tool for identifying and monitoring pediatric obesity [25,26,27]. According to our results, a high-fat diet was found to increase O-GlcNAcylation in many rodent and human tissues and regulate fatty acid synthesis, fat storage, and utilization [28,29]. Increased O-GlcNAcylation signaling prevents visceral fat lipolysis and promotes obesity in mice and humans [30], and excessive visceral fat accumulation is a primary risk factor for obesity and related diseases, including T2D [31]. Thus, we propose that increased O-GlcNAcylation might reveal early derangements in lipid metabolism and identify individuals at risk for T2D prior to overt alterations in glucose metabolism occurring. Consistent with our findings, whole blood protein O-GlcNAcylation was increased in healthy young adults with high homeostatic model assessment for insulin resistance (HOMA-IR) compared to subjects with low HOMA-IR, while no differences were observed in A1c between the two groups. Body fat, triglycerides, and insulin, although within the normal range, were significantly higher in the high-HOMA-IR group, again linking the O-GlcNAc status with subclinical metabolic alterations [32].

While the results are promising, this study is not without limitations. The small sample size represents a key limitation, reducing the statistical power and generalizability of the findings. However, this constraint is consistent with the exploratory nature of a pilot study, primarily intended to assess feasibility and generate preliminary data. Another limitation concerns the use of BMI as the sole anthropometric indicator to classify weight status, especially in pediatric subjects, where the body composition can vary significantly and quickly over time. Although widely used and validated in pediatric populations, BMI does not provide detailed information on body composition. A more detailed assessment of nutritional and metabolic status, including evaluation of the lean mass, fat mass, and muscle distribution, would be advisable in future studies.

Moreover, all participants were recruited from a single geographic area in Southern Italy, which may further limit the applicability of the results to other populations. Research involving pediatric subjects presents intrinsic challenges, particularly with regard to obtaining informed consent from legal guardians, a process that is often lengthy and logistically complex. These limitations emphasize the need for future studies that are ethically authorized, methodologically robust, and specifically designed to overcome the logistical and practical constraints typical of pediatric research. Multicenter studies will be needed to recruit larger cohorts of different ethnicities.

## 4. Materials and Methods

### 4.1. Enrollment of Pediatric Subjects

Pediatric subjects of Caucasian ethnicity aged 3–13 years (*n* = 8 normal weight and *n* = 8 overweight according to the WHO definitions [1]) were recruited over a period of 1 month at the Pediatric Operative Unit of Papardo General Hospital (Messina, Italy). After obtaining informed consent from the legal representatives, a peripheral blood specimen was collected from fasting study participants in the morning. These samples were tested in the Laboratory of Clinical Pathology Operative Unit (Papardo General Hospital, Messina, Italy). In parallel, a blood sample (~1.5 mL) for research was placed on ice and centrifuged, the plasma was discarded, and the pellet, including the erythrocytes and the buffy coat, was stored at −20 °C until further processing.

### 4.2. Determination of Protein O-GlcNAc Modification in Leukocytes by Western Blotting Analysis

After the elimination of erythrocytes (Red blood cell lysis solution, Promega, Milan, Italy) from the pellet, leukocytes were lysed on ice in a buffer containing 20 mM Tris-HCl, 150 mM NaCl, 1 mM EDTA, 0.1% NP40, and protease inhibitor cocktail (Roche, Basel, Switzerland). Western blotting was performed as described previously [33]. Shortly after, total proteins (10 μg) were separated on a 7.5% polyacrylamide gel and electroblotted on polyvinylidene fluoride membranes. Membranes were blocked for 1 h at room temperature in 5% bovine serum albumin and incubated overnight at 4 °C with a mouse monoclonal antibody (clone CTD110.6, diluted 1:1000, MABS1254, Sigma-Aldrich). Successively, membranes were incubated with peroxidase-conjugated goat anti-mouse IgG secondary antibodies (A9044, Sigma-Aldrich, Milan, Italy) diluted 1:10,000 in TBST solution for 1 h at 25 °C. To quantify the protein in equal amounts, a mouse monoclonal anti-β-actin antibody (A1978, Sigma-Aldrich, diluted 1:10,000) was incubated with the same membrane [34,35]. A system of chemiluminescence detection was employed to obtain the signal for image analysis. Densitometric analysis was performed using ImageJ software version 1.53 (National Institutes of Health, Bethesda, MD, USA).

### 4.3. Statistical Analysis

All data are expressed as arithmetic means ± standard errors of the mean. For statistical analysis and graphics, GraphPad Prism version 8.2 (GraphPad Software, Inc., San Diego, CA, USA) and Excel version 16.78.3 (Microsoft Corporation, Redmond, WA, USA) software were used. Statistical differences between two groups were verified by the two-tailed, unpaired Student’s *t*-test. Partial correlations were performed to assess the associations between leukocyte protein O-GlcNAcylation (residuals of X) and BMI, triglycerides, LDL-cholesterol, and CRP (residuals of Y), while controlling for potential confounding variables. Age and gender were included as covariates for blood parameters in all analyses. White blood cell count was included as a covariate for leukocyte O-GlcNAcylation levels. For each association, the Pearson correlation coefficient (r) and corresponding *p*-value are reported. Statistically significant differences were determined at * *p* < 0.05.

## 5. Conclusions

This study provides preliminary evidence that leukocyte protein O-GlcNAcylation is increased in overweight children compared to their normal-weight peers. This finding, observed in a pediatric population from Southern Italy, adds new insight into early molecular changes associated with excess weight in childhood. While studies exploring O-GlcNAcylation in adult populations across different ethnicities have been conducted, this is the first investigation focusing on pediatric subjects stratified by body weight, representing a novel contribution to the field. Elevated leukocyte O-GlcNAcylation in overweight children may serve as an early marker of metabolic imbalance and help identify individuals at higher risk of developing pre-diabetes or T2D, either during childhood or later in life. The observed associations with biochemical and anthropometric parameters support the potential role of O-GlcNAcylation in lipid metabolism and chronic inflammation. This study also highlights the value of integrating molecular markers with clinical and biochemical profiling to better understand the early pathophysiological mechanisms underlying metabolic disorders. The insights gained here lay the foundation for future longitudinal studies aimed at validating O-GlcNAcylation as a predictive biomarker of T2D and expand our understanding of pediatric metabolic health.

## Figures and Tables

**Figure 1 ijms-26-05665-f001:**
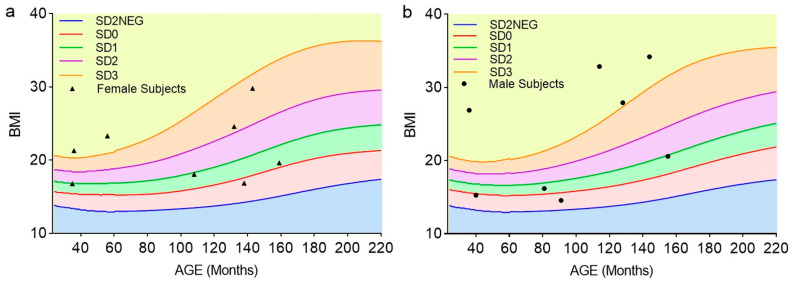
Graphical representation of BMI of all study participants plotted against z-score curves for (**a**) female and (**b**) male gender. According to WHO guidelines, growth curves allow children to be classified into percentile categories, providing a more accurate assessment of their nutritional status. Participants with BMI-for-age falling between −2SD and +1SD of the WHO Growth Reference median were classified as normal weight. Overweight was defined as a BMI +1SD above the WHO Growth Reference median (equivalent to a BMI of 25 kg/m^2^ for adults aged 19 years and older), while obesity was defined as a BMI +2SD above the WHO Growth Reference median (equivalent to a BMI of 30 kg/m^2^ for adults aged 19 years and older). (**c**) The O-GlcNAcylation of leukocyte proteins was probed by Western blotting, quantified by densitometry, and normalized to the housekeeping protein β-actin; data are from *n* = 8 normal-weight and *n* = 8 overweight participants. ** *p* < 0.01 compared to normal-weight participants, according to two-tailed, unpaired Student’s *t*-test. (**d**) Detection of global leukocyte protein O-GlcNAcylation in two normal-weight subjects (one male and one female, normal and overweight subjects, respectively) using Western blotting. Partial correlations between leukocyte protein O-GlcNAcylation and (**e**) BMI, (**f**) triglycerides, (**g**) LDL-cholesterol, and (**h**) CRP blood levels in overweight study participants (*n* = 8). Pearson’s correlation coefficient (r) and *p*-values are given.

**Table 1 ijms-26-05665-t001:** Clinical–biological parameters of normal-weight and overweight pediatric subjects divided by gender group. The physiological reference values are given according to the internal reference values of the Laboratory of Clinical Pathology Operative Unit of Papardo General Hospital (Messina, Italy).

Parameters	Normal-Weight Female Subjects (*n* = 4)	Normal-Weight Male Subjects (*n* = 4)	Overweight Female Subjects (*n* = 4)	Overweight Male Subjects (*n* = 4)	Physiological Reference Values
Age, months; average ± SD (range)	110 ± 4.61 (35–159)	91.75 ± 5.29 (40–155)	81.25 ± 6.21 (37–143)	105.5 ± 4.49 (36–144)	
Height, cm; average ± SD (range)	133.25 ± 2.61 (88–158)	131 ± 2.73 (104–170)	124.5 ± 1.65 (102–151)	131.75 ± 2.29 (91–150)	
Weight, Kg; average ± SD (range)	33.5 ± 2.42 (13–49)	31 ± 3.91 (16.5–59.5)	38.5 ± 2.51 (21–68)	64.62 ± 1.37 (55–83)	
FPG, mg/dL; average ± SD (range)	90.66 ± 0.94 (80–96)	85 ± 0.15 (84–86)	85.75 ± 1.10 (70–92)	82.4 ± 0.75 (70–86)	60–110 mg/dL
A1c, mmol/mol; average ± SD (range)	31.66 ± 0.10 (31–32)	34 ± 0.17 (33–35)	36.5 ± 0.93 (32–42)	37 ± 0.64 (32–42)	20–38 mmol/mol
Insulin, mlU/L; average ± SD (range)	7.03 ± 1.49 (3.7–13.3)	11.56 ± 5.20 (4.2–23.6)	8.85 ± 3.75 (2–22)	5.5 ± 1.02 (3–9)	2–25 mlU/L
Triglycerides, mg/dL; average ± SD (range)	85.75 ± 1.76 (66–102)	78.66 ± 2.14 (55–94)	137 ± 1.06 (120–150)	105.5 ± 0.79 (98–115)	30–180 mg/dL
Total Cholesterol, mg/dL; average ± SD (range)	141.25 ± 2.93 (122–190)	134.66 ± 1.29 (124–153)	182.5 ± 1.38 (156–196)	154 ± 2.74 (107–199)	0–190 mg/dL
HDL-Cholesterol, mg/dL; average ± SD (range)	53.5 ± 2.25 (33–74)	46.33 ± 1.15 (37–52)	46.75 ± 1.24 (38–55)	46 ± 0.95 (36–49)	>43 mg/dL
LDL-Cholesterol, mg/dL; average ± SD (range)	80.25 ± 2.36 (66–108)	78.33 ± 1.68 (61–93)	105.25 ± 1.44 (94–124)	81.25 ± 1.33 (64–92)	0–115 mg/dL
CRP, mg/dL; average ± SD (range)	0.28 ± 0.02 (0.27–0.29)	0.28 ± 0.02 (0.27–0.29)	3.60 ± 1.44 (1.84–6.43)	1.32 ± 0.03 (1.29–1.36)	0–0.5 mg/dL

FPG = Fasting Plasma Glucose; A1c = glycated hemoglobin; HDL = High-Density Lipoprotein; LDL = Low-Density Lipoprotein; CRP = C-reactive protein.

## Data Availability

The data that support the findings of this study are available from the corresponding author upon reasonable request.

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
