# Peer review of "Increased O-GlcNAcylation in Leukocytes from Overweight Pediatric Subjects: A Pilot Study"

_ijms, 2025, doi:10.3390/ijms26125665_

Round 1

Reviewer 1 Report

Comments and Suggestions for Authors
  1. In your title, you were written O-Glcnacylation like this, after that followed by the abstract it was mentioned O-GlcNAcylation like this. It should be modify uniformly.
  2. line 52 - T1D - abberiviation should be added in the first report.
  3. line 61 - its TD2 or T2D. Can you pleases check this. 
  4. After informed consent from the legal representatives, a peripheral blood specimen was collected from study participants and tested in the Laboratory of Clinical Pathology Operative Unit - Did you collect the blood before fasting or after food, it has any specific time?
  5. In parallel, a blood sample (~1.5 mL) for research was placed on ice and centrifuged, the supernatant was discarded, and the 98
    pellet, including the erythrocytes and the buffy coat, was stored at −20 °C. Here my question is the buffy coat was formed above the supernatant. How did you discard the supernatant?
  6. What secondary antibody was used in western blotting, if you used means the dilution facter and inclubating time should be include. 
  7. Western blotting image having lots of noise, what was the exact molecular weight of your target protein. it need to be notify in the picture.
  8. Your findings are being interpreted as overweight versus normal or overweight male versus overweight female.  Could you please elaborate? What about the normal group?
  9. Subtitle of conclussion should be added.

Author Response

REVIEWER 1  

In your title, you were written O-Glcnacylation like this, after that followed by the abstract it was mentioned O-GlcNAcylation like this. It should be modified uniformly. Done

Line 52 - T1D - abbreviation should be added in the first report. Done

Line 61 - its TD2 or T2D. Can you pleases check this. Done

After informed consent from the legal representatives, a peripheral blood specimen was collected from study participants and tested in the Laboratory of Clinical Pathology Operative Unit - Did you collect the blood before fasting or after food, it has any specific time?
Response: This information has been added to the text.

In parallel, a blood sample (~1.5 mL) for research was placed on ice and centrifuged, the supernatant was discarded, and the pellet, including the erythrocytes and the buffy coat, was stored at −20 °C. Here my question is the buffy coat was formed above the supernatant. How did you discard the supernatant?
Response: This sentence was rephrased to improve clarity.

What secondary antibody was used in western blotting, if you used to mean the dilution factor and incubating time should be included.
Response: This information has been added to the text.

Western blotting image having lots of noise, what was the exact molecular weight of your target protein. It needs to be notified in the picture.
Response: We want to thank the Reviewer for raising this point. Western blotting detected global O-GlcNAcylation of cellular proteins, not a single target protein. For this reason, the signal observed includes multiple bands, corresponding to different O-GlcNAc-modified proteins. The caption of Figure 1 has been partially revised to include this information, and the image has been improved in terms of resolution.

Your findings are being interpreted as overweight versus normal or overweight male versus overweight female?  Could you please elaborate? What about the normal group?
Response: We want to thank the Reviewer for raising this point. Clinical-metabolic parameter values belonging to pediatric normal-weight (female and male) subjects have been added to Table 1. All values from normal-weight and overweight participants fell within the physiological reference range, with the exception of CRP, which was above the reference range in both overweight female and male participants. Since our main objective was to investigate the potential differences in O-GlcNAc levels between pediatric normal-weight and overweight subjects, all comparisons in the study were performed between these two groups.

Subtitle of conclusion should be added. Done

Reviewer 2 Report

Comments and Suggestions for Authors

The paper: „Increased O-Glcnacylation Levels in Leukocytes from Overweight Pediatric Subjects: A Pilot Study” is an original study conducted on a pediatric population. The results obtained on pediatric population are always valuable and worthy of publication. However, since this study is conducted on very small group, it requires some modifications before it is considered for publication.

  1. Since O-Glcnacylation is a posttranslational process, it is unusual to say O-Glcnacylation Levels. It is more appropriate to just say “Increased O-Glcnacylation in Leukocytes” or “Increased O-GlcNAc levels”. Please change the title and this phrase throughout the text including the x-axis on the Figure 1.
  2. There is a need for the Clinical-biological parameters of all or normal weight vs overweight pediatric subjects divided by gender groups (or in total) given as Table 1 in Material and Methods, before any statistical analysis is conducted. So, we need to see the characteristics of all the subjects involved in the study.
  3. The sensitivity and specificity of the test(s) should be given, if possible, especially having in mind the limited number of participants in order to eliminate the false positive and the false negative results.
  4. Limitations and strengths of the study are much needed before the conclusion. First, the number of the participants is very limited and may not be enough for statistical significance. Based on the standard deviation of the O-Glcnacylation Levels in the pediatric population, the required minimum of participants in a study in order to achieve statistical significance is much higher. This should be emphasized. Furthermore, this is monocentric study and the obtained results may not be easily transferable to other ethnicities or races since there is no data regarding the variations in O-Glcnacylation Levels among races. If there is such data, please address this issue in the discussion part. The study has been conducted on children in the age between 3 and 13, thus the obtained results may not be extrapolated to younger or older age especially given the changes in the fraction of leucocyte types before the age of 5. The strengths of this paper should also be put in focus before the conclusion part. This study has original approach in evaluating O-Glcnacylation Levels in overweight children. Underline the strengths and the originality of the idea presented in this research although it is a pilot study.

Author Response

REVIEWER 2

The paper: “Increased O-Glcnacylation Levels in Leukocytes from Overweight Pediatric Subjects: A Pilot Study” is an original study conducted on a pediatric population. The results obtained on pediatric population are always valuable and worthy of publication. However, since this study is conducted on very small group, it requires some modifications before it is considered for publication.
Response: We thank the Reviewer for the overall positive evaluation.

Since O-Glcnacylation is a posttranslational process, it is unusual to say O-Glcnacylation Levels. It is more appropriate to just say “Increased O-Glcnacylation in Leukocytes” or “Increased O-GlcNAc levels”. Please change the title and this phrase throughout the text including the x-axis on the Figure 1. Done.

There is a need for the clinical-biological parameters of all or normal weight vs overweight pediatric subjects divided by gender groups (or in total) given as Table 1 in Material and Methods, before any statistical analysis is conducted. So, we need to see the characteristics of all the subjects involved in the study.
Response: We want to thank the Reviewer for raising this point. In accordance with the suggestion of this Reviewer and Reviewer #1, Table 1 has been revised to include the clinical-biological parameters of normal-weight participants, allowing a more complete comparison between normal-weight and overweight pediatric subjects. All values from normal-weight and overweight participants fell within the physiological reference range, with the exception of CRP, which was above the reference range in both overweight female and male participants.

The sensitivity and specificity of the test(s) should be given, if possible, especially having in mind the limited number of participants in order to eliminate the false positive and the false negative results.
Response: We want to thank the Reviewer for raising this point. We assume that the Reviewer refers to the sensitivity and specificity of the O-GlcNAcylation test. However, since this is a pilot study, there is no possibility of calculating these parameters. Instead, our main objectives were to demonstrate the feasibility of the study, to obtain preliminary data on the potential differences in O-GlcNAc levels between pediatric normal-weight and overweight subjects, and to assess the variability of leukocyte protein O-GlcNAcylation in overweight pediatric subjects to calculate the minimum number of participants to be recruited in the follow-up full-scale study. The scope of the study is now better explained in the revised manuscript at the end of the introduction section.

Limitations and strengths of the study are much needed before the conclusion. First, the number of the participants is very limited and may not be enough for statistical significance. Based on the standard deviation of the O-Glcnacylation Levels in the pediatric population, the required minimum of participants in a study in order to achieve statistical significance is much higher. This should be emphasized. Furthermore, this is monocentric study and the obtained results may not be easily transferable to other ethnicities or races since there is no data regarding the variations in O-Glcnacylation Levels among races. If there is such data, please address this issue in the discussion part. The study has been conducted on children in the age between 3 and 13, thus the obtained results may not be extrapolated to younger or older age especially given the changes in the fraction of leucocyte types before the age of 5. The strengths of this paper should also be put in focus before the conclusion part. This study has original approach in evaluating O-Glcnacylation Levels in overweight children. Underline the strengths and the originality of the idea presented in this research although it is a pilot study.
Response: In response, we have added a dedicated paragraph discussing the limitations and strengths of our study in the conclusion section. In particular, regarding the sample size, we acknowledge that the number of participants is limited and may affect the statistical power of the study. We have explicitly stated this limitation in the manuscript. Moreover, we emphasize that this is a pilot study, and as such, it was designed to include a relatively limited number of subjects, to demonstrate feasibility and calculate the minimal number of participants to be recruited for a future larger study. This is now better explained in the manuscript (objective, discussion, and conclusions). Based on the standard deviation of the O-GlcNAcylation in the pediatric overweight population, we have calculated a n=17 for a full-scale study. This is now stated in the Results section. Nonetheless, despite the limited sample size, the observed difference in O-GlcNAc levels between normal-weight and overweight subjects was evident and statistically significant, supporting the reliability of the result. Moreover, the study intentionally targeted the pediatric population aged 3 to 13 years, a range that falls within the World Health Organization’s definition of childhood.

As for the monocentric nature of the study, we have now clearly stated in the manuscript (end of introduction and conclusions) that the study was conducted at a single center; however, we feel that the data could be representative of other populations. In this regard, it is important to note that previous studies have already investigated O-GlcNAcylation in adult populations from different geographical and ethnic backgrounds. For example, one study conducted on adult volunteers from the Western Cape region of South Africa, classified as normoglycemic, prediabetic, or diabetic, reported a cell-type specific modulation of O-GlcNAcylation levels. In particular, increased O-GlcNAc levels were observed in lymphocytes of diabetic individuals and in granulocytes of prediabetic subjects, when stratified by fasting plasma glucose. These findings suggest a nuanced immune-metabolic response associated with glycemic status (DOI: 10.1210/jc.2012-2229). These data, although obtained in adults stratified by glycemic status and not body weight, demonstrate that studies on O-GlcNAc biology are being extended to diverse populations; however, their applicability to pediatric settings still needs to be validated, which reinforces the relevance and novelty of our current study.

Reviewer 3 Report

Comments and Suggestions for Authors

This study has examined the association between leukocyte protein O-GlcNAcylation in a cohort of overweight children from South Italy. But there are some issues must be addressed.

1) the sample size of this study is too small; the power is relatively low. A lot of confounders were not considered.

2) when the authors analyzed the correlation between O-GlcNAcylation levels and BMI and other indicators, some important confounders were not considered, including age, sex, leukocyte counts were not considered.

3) one of important comparisons should be the baseline characteristics by overweight status. But the study did not show this.

4) the young participants’ BMI changed so fast, even the BMI values could be indicative of totally different body statues. I do not think these BMI values can be used to be fitted across those study subjects.

Minor comments:

  • “rbitraryunit” in Figure 1e should be revised.
  • “Partecipants” in Figure 1c should be revised.
  • “A positive correlation between data sets was tested by linear regression and expressed as Pearson’s r coefficient.” Correlation cannot be tested by linear regression and it should be tested by partial correlation analyses.

Author Response

REVIEWER 3

This study has examined the association between leukocyte protein O-GlcNAcylation in a cohort of overweight children from South Italy. But there are some issues must be addressed.

  1. The sample size of this study is too small; the power is relatively low. A lot of confounders were not considered.
    Response: We want to thank the Reviewer for raising this point. We are conscious of the limitations associated with the relatively small sample size. Data obtained should be interpreted with caution and confirmed by larger studies. Nevertheless, we believe that the data collected provides useful preliminary indications, as further discussed below.

  2. When the authors analyzed the correlation between O-GlcNAcylation levels and BMI and other indicators, some important confounders were not considered, including age, sex, leukocyte counts were not considered.
    Response: We thank the Reviewer for this valuable observation. We have now addressed this concern by including age, sex, and leukocyte counts in our analysis. Figure 1 has been modified to add a panel showing partial correlation analysis between leukocyte protein O-GlcNAcylation and BMI, controlling for confounding factors including gender, age, and white blood cell count.

  3. One of important comparisons should be the baseline characteristics by overweight status. But the study did not show this.
    Response: We want to thank the Reviewer for raising this point. In accordance with the suggestions of all three Reviewers, we have revised Table 1 to include a comparison of baseline clinical and biological characteristics between normal-weight and overweight pediatric subjects.

  4. The young participant BMI changed so fast, even the BMI values could be indicative of totally different body statues. I do not think these BMI values can be used to be fitted across those study subjects.
    Response: We want to thank the Reviewer for raising this point. We recognize that during childhood, the body undergoes rapid changes in terms of body composition, height, and fat distribution, which affect the interpretation of BMI. However, we provide some considerations below to support our approach. BMI is a widely used tool for assessing weight status in children due to its simplicity and rapid measurement. Although it has limitations, such as its failure to distinguish between fat mass and lean mass, BMI is a useful epidemiological marker. Several studies have shown that BMI well correlates with fat mass in children, providing a valid tool for identifying and monitoring pediatric obesity [DOI:10.1542/peds.2008-3586E; DOI: 10.1111/ijpo.242; 10.1016/j.numecd.2009.08.008]. These considerations have been added to the discussion sections, and these articles were referenced. Moreover, BMI values in children are interpreted using age- and sex-specific growth curves, such as those provided by the World Health Organization (WHO). Growth curves allow children to be classified into percentile categories, providing a more accurate assessment of their nutritional status. This study has a cross-sectional design, in which BMI measured at a single time point is used as an indicator of weight status. While recognizing that BMI may not accurately reflect weight change over time, we believe that the collected data offer some useful preliminary insights, especially for future hypotheses to be tested in longitudinal studies. A sentence about this issue has been added to the full text.

Minor comments:

  • “rbitraryunit” in Figure 1e should be revised. Done.
  • “Partecipants” in Figure 1c should be revised. Done.
  • “A positive correlation between data sets was tested by linear regression and expressed as Pearson’s r coefficient.” Correlation cannot be tested by linear regression and it should be tested by partial correlation analyses. Done. See figure 1, panel i.

Round 2

Reviewer 1 Report

Comments and Suggestions for Authors

Dear Authors, 

         Really appreciate your efforts

Author Response

Dear Reviewer,
we truly appreciate your constructive comments, which have been very helpful in improving the quality of our manuscript. Thank you for your time and valuable feedback.

Reviewer 3 Report

Comments and Suggestions for Authors

This study should not consider uncorrected BMI as the outcome for children, since the uncorrected BMI cannot be compared for children whose BMI cannot be compared directly. In addition, this study is of small sample size. I noticed that the participants with higher levels of TG, LDL, CRP would be likely to be female overweight. And the authors only selected 8 participants in the correlation analyses, while the total participants were 16 participants. It is totally unsuitable. In addition, the statistical methods is not suitable in this study. In the female overweight group, the weights are 124.1.65 ± 1.23, but the range is 120–157. It must lead to statistical issues.

Author Response

A point-by-point response to the comments is attached.

Round 3

Reviewer 3 Report

Comments and Suggestions for Authors

"A partial correlation analysis accounting for confounders (gender and age for meta- 110
bolic/inflammatory parameters, and white blood cell count for O-GlcNAcylation) re- 111
vealed significant association between leukocyte protein O-GlcNAcylation and BMI (Fig- 112
ure 1e), lipid metabolism markers (triglycerides and LDL-cholesterol respectively, Figure 113
1f,g), as well as the inflammatory marker CRP (Figure 1h)." but the response saying that "selected 8 participants in overweight participants". The analyses were done for overweight participants. But the result description is for all participants.

BMI cannot be compared across these participants.

The sample size is too small. So results is not solid!
